# Tetraspanins: Host Factors in Viral Infections

**DOI:** 10.3390/ijms222111609

**Published:** 2021-10-27

**Authors:** ChihSheng New, Zhao-Yong Lee, Kai Sen Tan, Amanda Huee-Ping Wong, De Yun Wang, Thai Tran

**Affiliations:** 1Infectious Disease Translational Research Program, National University of Singapore, Singapore 119228, Singapore; phsncs@nus.edu.sg (C.N.); leezhaoyong21@u.nus.edu (Z.-Y.L.); kaistan@nus.edu.sg (K.S.T.); phswhpa@nus.edu.sg (A.H.-P.W.); 2Department of Physiology, Yong Loo Lin School of Medicine, National University of Singapore, Singapore 117593, Singapore; 3Department of Microbiology and Immunology, Yong Loo Lin School of Medicine, National University of Singapore, Singapore 117545, Singapore; 4Department of Otolaryngology, Yong Loo Lin School of Medicine, National University of Singapore, Singapore 119228, Singapore; 5Biosafety Level 3 Core Facility, Yong Loo Lin School of Medicine, National University Health System, National University of Singapore, Singapore 119228, Singapore

**Keywords:** CD9, CD63, CD151, CD81, coronavirus, COVID-19, HIV, HPV, influenza, tetraspanin, virus, Zika

## Abstract

Tetraspanins are transmembrane glycoproteins that have been shown increasing interest as host factors in infectious diseases. In particular, they were implicated in the pathogenesis of both non-enveloped (human papillomavirus (HPV)) and enveloped (human immunodeficiency virus (HIV), Zika, influenza A virus, (IAV), and coronavirus) viruses through multiple stages of infection, from the initial cell membrane attachment to the syncytium formation and viral particle release. However, the mechanisms by which different tetraspanins mediate their effects vary. This review aimed to compare and contrast the role of tetraspanins in the life cycles of HPV, HIV, Zika, IAV, and coronavirus viruses, which cause the most significant health and economic burdens to society. In doing so, a better understanding of the relative contribution of tetraspanins in virus infection will allow for a more targeted approach in the treatment of these diseases.

## 1. Introduction

Tetraspanins belong to a family of transmembrane glycoproteins, with 33 members being identified in humans. In particular, CD9, CD63, CD81, CD82, CD151, and TSPAN7 (CD231) were implicated in numerous chronic and infectious disease pathologies [1,2]. Structurally, tetraspanins consist of four transmembrane regions (designated as TM1 to TM4) and two extracellular loops (EC1 and EC2), with a variable EC2 region that contains specific sites to facilitate interactions with partner proteins [1]. Owing to the hydrophobic TM domains, tetraspanins can establish unique platforms at the cell surface, known as tetraspanin-enriched microdomains (TEMs) [1,3], allowing for lateral associations with partner proteins, including laminin-binding integrins, adhesion molecules, growth factor receptors, metalloproteinases, intracellular signaling molecules, and even other tetraspanin family members. Through their interactions with TEMs, the function of associated molecules is modulated, stabilized, or even prevented [4,5,6]. Thus, canonically, tetraspanins were found to regulate cellular migration, adhesion, fusion, signaling, and even metastasis [7,8,9]. With the current impact of SARS-CoV-2 and its variants, there is an increasing need for a deeper understanding of host–pathogen interactions. Given the broad tissue distribution, expression, and mediation of numerous physiological processes [10], it is unsurprising that viruses hijack tetraspanins for their life cycle at sites of receptor binding, endocytosis, trafficking, entry, viral replication, nuclear export, and viral budding. In support of this, successful targeting of tetraspanin CD81 was observed for the treatment of the hepatitis C virus (HCV) [2,11]. In this review, we aimed to provide an update on the role of tetraspanins in the replication of various viral pathogens that have caused the most significant health and economic burden to human society in recent years. These include the influenza virus (IAV), human immunodeficiency virus (HIV), human papillomavirus (HPV), Zika virus (ZIKV), and coronavirus (CoV).

## 2. Human Papillomavirus (HPV)

Globally, cervical cancer is the fourth most common cancer and the fourth leading cause of cancer death in women [12]. Various HPVs were shown to cause cervical cancer, most commonly subtypes HPV16 and HPV18 [13]. Other benign HPV subtypes, such as HPV6 and HPV11, cause cutaneous warts on the hands and feet. The risk of cervical cancer can be significantly reduced by undergoing regular screening for early detection, such as a Papanicolaou (Pap) test and HPV testing. More than 90% of HPV-attributable cancers can be potentially prevented with the current HPV vaccines available [14]. However, the underlying molecular mechanisms of viral replication, particularly the role of tetraspanins, remain to be fully defined. The HPV infectivity cycle relies on a multistep process that involves various tetraspanins at different stages: CD151 and CD9 at the viral entry (endocytosis) stage and CD63 (post-endocytosis) during trafficking.

### 2.1. Virus Entry

HPV is a small, non-enveloped, double-stranded DNA virus of about 8000 base pairs in size [15]. Its genome is wrapped by histones and encapsulated by two capsid proteins, namely, L1, the major capsid protein, and L2, the minor capsid protein [16]. Both capsid proteins play important roles in initiating virus binding to the host cells’ receptors. Like most viruses, HPV invades and hijacks the host cell’s replication machinery to replicate and propagate itself. HPV is an opportunistic virus that enters the basal layer of dividing keratinocytes through micro-abrasions. Upon entering the host cells, the virus binds to heparin sulfate proteoglycans (HSPG) or non-HSPG components, such as laminin-332 [17,18] (Figure 1). This initial binding step of HPV was first reported by Joyce et al., who showed that HPV11 binds to HPSG in vitro using immortalized human keratinocytes namely, HaCaT cells [18]. Furthermore, Giroglou et al. showed that other common HPV subtypes, such as HPV16 and HPV33, also use HPSG as a host-binding receptor, thus solidifying HPSG as a receptor for HPV [4]. Upon binding to HPSG, the viral particle undergoes conformational changes and is transported to a secondary binding complex. However, the components of this complex are not well characterized. Various studies have shown that α6 integrin, growth factor receptors, and tetraspanins all play important roles in the endocytosis of the virus [19,20,21]. Scheffer et al. reported that CD151 involvement in HPV16 infection is not at the initial binding to the HPSG receptor, but it is required for the endocytosis of the virions [22]. Through elegant immunofluorescence studies, Wüstenhagen et al. showed that obscurin like-1 (OBSL1), a cytoskeletal adaptor protein, also colocalizes with the CD151 and HPV16 capsid protein L2 for endocytosis into target cells [23].

Another tetraspanin that was shown to play a role in HPV signaling is CD9, which appears to affect the post-binding steps. Changes in the magnitude of CD9 expression levels are correlated with HPV infection rates, with low CD9 expression levels supporting infection and the promotion of ADAM17 sheddase activity and ADAM17-mediated ERK phosphorylation; these last two processes are required for the assembly of the HPV entry platform [24].

### 2.2. Virus Trafficking

Upon viral entry, the endocytosed HPV trafficks into endosomes for sorting into various locations. Immunofluorescence and electron microscopy studies of HPV (HPV16, HPV18, HPV31)-infected epithelial (HeLa, HaCaT, and NHEK) cells showed that CD63 forms a complex with syntenin-1, aiding the post-endocytosis trafficking of the virus to multivesicular endosomes [20,25]. CD63 siRNA-treated HeLa, HaCaT, and primary keratinocytes exhibited decreased infection by more than 50%, while rescuing CD63 was sufficient to re-establish the viral infectivity rate [25]. Following this, multivesicular endosome capsids disassemble, and the L1 capsid is degraded by lysosomes. At the same time, the rest of the viral DNA is uncoated, and, together with L2, the complex is transported into the nucleus via the trans-Golgi network system [26].

### 2.3. Nuclear Entry and Replication

Not much is known about the nuclear entry of HPV other than that the viral DNA and L2 enter the nucleus as a complex [27]. Whether tetraspanins are involved in mediating HPV nuclear entry remains to be determined. As for viral replication, HPV genes rely on the cell cycle machinery of host cells, highjacking the host cell’s prophase cell cycle step. Additionally, only the cells from the basal layer of the stratified epithelial cells progress through this step [28]. It appears that HPV resides at the trans-Golgi network of cells in the interphase and only enters the cells when it reaches the prophase. Like most DNA viruses, HPV separates itself from the host cell cycle stage to optimize its replication. An interesting finding from Reinson et al. showed that the HPV utilizes its viral replication proteins E1 and E2 to initiate its replication in the host cell S phase and extend its replication into the G2 phase to increase its copy numbers [29].

## 3. Human Immunodeficiency Virus (HIV)

HIV is an enveloped, single-stranded, positive-sense RNA virus belonging to the *Retroviridae* family’s *Lentivirus* genus [30,31]. HIV is further classified into two types, HIV type-1 (HIV-1) and HIV type-2 (HIV-2). Globally, HIV-1 is the more common type in circulation [32,33], with over 37 million individuals living with HIV-1 worldwide [34]. Moreover, HIV is the causative agent in the pathogenesis of acquired immunodeficiency syndrome (AIDS). By targeting CD4+ T cells [35], macrophages [36], and dendritic cells (DCs) [37], HIV gradually weakens the human immune system over 10–15 years in the absence of clinical interventions. This period allows for potentially lethal opportunistic infections of tuberculosis, pneumonia, herpes simplex, Kaposi’s sarcoma, and coccidioidomycosis secondary to AIDS [38].

Similar to most viral pathogens, vaccines remain an attractive yet elusive option to control the spread of HIV [39]. The main challenge of HIV vaccine development is the presence of only one surface viral antigenic target, namely, the trimeric envelope (Env) spike protein. Structurally, Env glycoproteins have a median of 30 N-linked glycosylation sites per protomer on their ectodomain. This grants the Env protein a substantial glycan mass (a ‘glycan shield’) that covers its surface, preventing access and thus recognition of Env by humoral immune responses [40,41]. Coupled with the highly mutagenic nature of the HIV RNA genome [39], vaccine development for circulating strains of HIV remains a monolithic task. To date, only the RV144 HIV vaccine trial in Thailand has demonstrated modest results of 31% vaccine efficacy [42,43]. Although noteworthy vaccine candidates are still currently in development (e.g., eOD-GT8 60mer) [44], an efficacious prophylactic HIV vaccine remains out of reach.

The current standard of care for HIV-1 patients involves antiretroviral therapy (ART), which was shown to effectively suppress viral loads and reduce mortality [37]. However, ARTs cannot eradicate HIV from infected individuals, allowing for rebounds in viral loads due to the release of virions from ART-insensitive cells that harbor latent proviruses [45,46]. Hence, a major limitation of ARTs is the suboptimal adherence that is observed in patients [47]. Thus, it is crucial to identify and characterize both the host and viral protein targets that would aid the development of antivirals that are capable of eliminating active and/or latent HIV particles.

### 3.1. Virus Entry

To gain access to its target cells, HIV-1 has adopted different strategies that are uniquely regulated by tetraspanins (Figure 2). Intriguingly, where human DCs are poor hosts for HIV-1 replication, they can uniquely present virions to T lymphocytes that are more permissive to infection [48]. This is attributed to the actin network that is necessary for the formation of DC dendrites. Due to the vast cortical actin networks present, high intracellular tensions are generated, inhibiting endocytosis in general [49,50]. The tetraspanin TSPAN7, through either direct or indirect association with ARP2/3, positively regulates actin nucleation and thus actin polymerization [51]. Accordingly, TSPAN7 is also associated with the extracellular retention of HIV-1 virions at actin-rich dendrites, while TSPAN7 knockdown results in the internalization of HIV-1 virions via macropinocytosis in DCs [51]. Although actin polymerization was established to be vital for Env-mediated entry into T cells [52], the reverse appears to be true for dendritic cells (DCs). Hence, DCs serve as couriers for HIV-1 when TSPAN7 is present. HIV-1-bound dendrites are presented to T lymphocytes, which are more susceptible to infection, thus enhancing the spread of HIV-1 within the host. HIV-1′s exploitation of the host exosomal machinery was well-established [53,54]. HIV-1 can bind to exosomal protein TIM-4, gaining entry into the cell via exosome-related forms of endocytosis [55]. The blocking of tetraspanins CD9 and CD81 on exosomes limited the exosomal endocytosis in DCs, and consequently, caused a drop in HIV-1 infection [56].

The HIV-1 spike protein Env facilitates the entry of the virus into host cells. Host CD4 receptors interact with the Gp120 subunit of Env, inducing a conformational change in gp120. This allows for the binding of host co-receptors CCR5 or CXCR4, initiating endocytosis [30].

Tetraspanins were also shown to be intricately involved in host–receptor regulation, modulating receptor accessibility and thus viral entry into target cells. In T lymphocytic cells, CD81 knockdown enhanced both HIV viral syncytia formation and Env-mediated endocytosis [57]. Conversely, the overexpression of CD81 limited viral entry [57]. CD9 knockdown and overexpression produced similar patterns in the entry of HIV-1 and may be critical in the infection of macrophages [57,58]. CD4 also interacts with CD81 [59,60,61], providing a platform for CD4 homodimerization at TEMs [61]. Although the role of CD4 dimers in HIV-1 replication remains unclear, it is suggested to limit the accessibility of CD4 receptors to gp120, and thus restrict viral entry [30,61]. Separately, the tetraspanin CD63 was also shown to regulate CXCR4 expression on T cell plasma membranes, thereby limiting HIV-1 viral entry [62]. CD63′s N-linked glycans interact with CXCR4 at the Golgi apparatus [63]. This interaction alters the trajectory of CXCR4 from the plasma membrane to the late endosome or lysosomes, thereby decreasing receptor availability at the cell surface and causing T lymphocytes to be less permissive to infection [62,63]. Interestingly, CD63 knockdown is associated with lowered HIV-1 viral titers in macrophages [64]. This was observed in lab-adapted R5- and R5X4-tropic HIV-1 strains, suggesting that CD63 is specifically involved in viral entry pathways that are facilitated by the co-receptor CCR5 [64]. Hence by altering host receptor interactions and localization, tetraspanins control receptor accessibility, rendering cells as being less permissive to viral entry.

The spread of HIV-1 may also occur directly across a virological synapse, where the donor may present the HIV-1 virion to a recipient cell, resulting in the infection of the recipient [65]. This increases the accessibility of target cells to HIV-1 and reduces the duration in which HIV-1 exists exogenously, thus lowering the risk of immune detection and elimination [65]. Tetraspanins CD63, CD81, and CD9 are recruited to virological synapses between T cells, and their depletion is associated with a drop in synapse formation [66]. More specifically, the large extracellular loop of CD63 was shown to interact with the HIV-1 gp41 protein [67]. This is postulated to prevent Env-mediated fusion between the donor and recipient cells, thus preventing syncytia formation [68]. Since multinucleation typically results in the activation of apoptotic pathways [69,70], the infection of an apoptotic cell would be unproductive. Hence, the maintenance of the virological synapse by CD63 would lower the risk of fusion activities, increasing the potential of successful HIV-1 infections.

### 3.2. Transcription/Replication

Once HIV-1 enters the target cell, the RNA genome is released, and reverse transcription occurs, ultimately generating DNA [71]. The viral DNA moves into the nucleus and is integrated into the host genome by viral integrase [71]. Once fully integrated, HIV-1 is considered a provirus. Viral mRNA is expressed with the aid of the viral trans-activator of transcription (Tat) protein and several other host factors. Viral mRNA is used for the transcription of other viral proteins (e.g., gp120, gp41, negative regulatory factor (Nef), viral protein U (Vpu), and group-specific antigen (Gag)). If the full length of the viral mRNA is expressed, it is eventually packaged into a progeny virus as the viral genome [71].

Although the function of tetraspanins at the plasma membrane was extensively studied, tetraspanins also modulate intracellular signaling and trafficking events [2]. Unsurprisingly, tetraspanins were also shown to regulate the intracellular aspects of HIV-1′s replication cycle. In T lymphoblasts and HELA cells, CD81 was shown to directly bind host deoxynucleotide triphosphate phosphohydrolase SAMHD1, promoting the proteasomal degradation of SAMHD1. SAMHD1 degrades deoxynucleoside triphosphates (dNTP), limiting substrate dNTP levels in the cytoplasm. Therefore, by decreasing the SAMHD1 protein abundance, CD81 ensures sufficient cytoplasmic dNTP substrate for reverse transcription of HIV-1 RNA [72]. Independently, CD63 siRNA (small interfering RNA) knockdown is correlated with lowered HIV-1 virus titers in macrophages [73], T lymphocytes [64,74], and DC [74] culture supernatants. This depletion in CD63 seemed to affect the initiation and completion of HIV-1 reverse transcription, integration of HIV-1 DNA into the host genome, and the production of the early HIV protein Tat [74,75]. Since Tat modulates the expression of other HIV-1 proteins, this reduction in Tat activity was met with an expected decrease in the production of a late HIV-1 protein antigen, p24 [74,75]. Taken together, current evidence suggests that tetraspanins support the early phases of HIV-1′s replication cycle in target immune cells. Hence, lowering tetraspanin CD81 and/or CD63 expression or blocking their activity during early infection seems to be a promising strategy to limit reverse transcription, genomic integration, and ultimately viral replication.

### 3.3. Assembly and 3.4 Budding/Egress

Currently, the assembly of HIV-1 virions remains largely controversial, with research distributed between two different models: the spontaneous/self-assembly model or the host-catalyzed model [76,77]. Agreement between the two models lies with HIV-1 polyprotein Gag and Gag-RNA interactions driving virion assembly [76].

Similar to many different envelope viruses, HIV-1 assembles at CD9-, CD63-, CD81-, and CD82-containing TEMs [66,78,79]. Individually, Env and Gag proteins were also shown to colocalize with CD9, CD63, and CD81 at the host T cell plasma membrane, with Gag directly interacting with CD81 [66,79]. In particular, CD63 is trafficked from intracellular compartments to HIV-1 assembly sites in macrophages [80]. However, CD63 knockdown was not associated with a reduction in viral release, and HIV-1 virions could still assemble at CD81- and CD9-containing regions [80]. This implies that there may be redundancies or compensatory mechanisms in tetraspanin functions, and a more general suppression of tetraspanins may be required to hamper later phases in HIV replication.

Additionally, during this phase of the replication cycle, HIV-1 proteins can also coordinate the expression and localization of tetraspanins. In HIV-1-infected T lymphocytes, a total decrease in the expression of CD81 (80%) and CD82 (62%) was observed [81,82]. This downregulation was attributed to the HIV-1 viral protein Vpu, which binds directly to CD81, enhancing proteasomal and lysosomal degradation [82]. Consequently, CD81 expression at the plasma membrane is diminished. This was carried forth to the resultant HIV-1 progeny, where low CD81 levels were observed in the viral envelopes [82]. Low CD81 levels in the HIV-1 envelope are also associated with an enhanced viral titer in subsequent T cell infections, potentially due to an enhanced ability for viral entry [82]. A similar trend is observed with CD63 in Molt4/IIIB T cells, whereupon phytohemagglutinin (PHA) and phorbol 12-myristate 13-acetate (PMA) activation occurred, cell surface CD63 expression decreased, and the presence of CD63 on progeny viral envelopes was also limited [83]. CD63 on progeny HIV-1 surface was also found to interfere with infection, and this interference was extended to other tetraspanins (CD9, CD81, CD82, and TSPAN7) [83]. In HIV-1-infected human monocyte-derived macrophages, CD81, CD82, CD9, TSPAN14, CD53, and CD63 were present on highly purified HIV-1 progeny virions [84]. However, whether the presence of tetraspanins interferes with HIV-1 viral infection remains to be determined.

More recently, it was found that viral proteins Vpu and Nef are individually capable of altering the expression of a wide array of tetraspanins (including CD81, CD63, CD151, TSPAN7, CD53, and CD37) by over 35%, causing the enrichment of these tetraspanins at the perinuclear area of T cells [81]. Another potential regulatory mechanism involves the small GTP-binding protein Rab3a. Functionally, Rab3a activity is associated with exocytosis [85] and CD63 lysosomal and proteasomal degradation [86]. Although more studies are needed, HIV-1 may hijack the activity of the host Rab3a protein to induce CD63 degradation and, thus, enhance viral infectivity of the progeny. Much is still unknown about the intricate coordination of tetraspanins during HIV-1 viral assembly. Furthermore, the control of tetraspanins at cellular surfaces seems to be cell-type specific, adding complexity to the replication cycle of HIV-1.

To further evade immune detection and eradication, HIV-1 hijacks host exosomal machinery exosomes that are derived from HIV-1 T cells, and infected immature DCs were found to be capable of infecting T cells in the absence of HIV-1 virion [53,87]. These exosomes from HIV-infected immature DCs were found to contain a combination of both HIV-1 and host molecules, where HIV’s ability to infect is closely associated with the presence of host fibronectin and galectin-3 [88,89]. It was established that tetraspanins are extensively involved in the formation and regulation of exosomes. Indeed, as the CD63, CD81, and CD9 expressions in HIV-1-infected T cells were enhanced for up to 24 h post infection, infectious exosomes displayed higher CD81 and CD9 [87]. Whether the presence of tetraspanins enhances exosome-mediated infections remains to be determined. Separately, despite being non-infectious themselves, exosomes released 3 h post infection were capable of priming and thus enhancing the infection of recipient cells by HIV-1 virions [87].

The role of tetraspanins is observed in almost all facets of HIV-1′s replication cycle. However, the regulation and interactions between tetraspanins, other host proteins, and viral proteins are highly nuanced and dependent on the HIV-1 replication stage. Furthermore, due to multiple target cell types and the crosstalk between these cells, clinical interventions that target tetraspanins would need to be highly specific.

## 4. Zika Virus (ZIKV)

ZIKV is an enveloped, non-segmented, single-stranded, positive-sense RNA virus belonging to the family *Flaviviridae*. It is closely related to the dengue virus, West Nile virus, Japanese encephalitis virus, and yellow fever virus [90,91,92,93]. Where the ancestral African ZIKV pathogenicity is almost exclusively enzootic, the Asian lineage and its American subclade of ZIKV are capable of transmission via human-adapted *Aedes* mosquitoes [91,94]. ZIKV infections typically result in acute febrile illness with symptoms of rashes, fever, arthritis or arthralgia, conjunctivitis, myalgia, headache, retro-orbital pain, edema, and vomiting [94]. Though hospitalizations for these symptoms are rare, recent pandemics (Micronesia 2007, French Polynesia 2013–2014) revealed complications arising from ZIKV infections that were unexpected and severe. Namely, neurological complications observed as Guillain–Barré syndrome in adults, and congenital Zika syndrome (CZS) observed as microcephaly, congenital malformation, and potential fetal demise in about 6–11% of infected pregnant women [92,94]. Owing to its severe neurologic and teratogenic effects, the World Health Organization declared ZIKV to be a public health emergency of international concern in February 2016, predicting that it will remain a public health challenge in the ecological niches of the *Aedes* mosquito [92,95]. Current research into ZIKV therapeutics is limited and largely in the early development phases; antiviral and antibody-based therapeutics for ZIKV are in preclinical stages, while only a single vaccine candidate is under active phase 2 tests [95]. Hence, it is imperative to understand further the molecular mechanisms of ZIKV replication and its ability for prenatal transmissions to identify potential host and/or viral drug targets.

### 4.1. Virus Entry

Current evidence for the exact receptors that are involved in ZIKV entry remains controversial [90]. However, due to its ability to modulate the entry of other flaviviruses (West Nile virus and dengue virus) [96], Human T-cell immunoglobulin and mucin-domain containing proteins (TIM1) were asserted to potentially modulate ZIKV entry as well [90]. Upon receptor recognition and binding, ZIKV virus uptake involves clathrin-dependent endocytosis, endosomal acidification, the fusion of the endosomal membrane with the viral membrane, and the release of the ZIKV genomic RNA into the cytoplasm [90] (Figure 3).

### 4.2. Transcription/Replication

Upon release, the single-stranded, positive-sense RNA genome of ZIKV is used as a template for synthesizing its complementary negative-sense RNA using the viral non-structural protein 5 (NS5). NS5 has RNA-dependent RNA polymerase (RdRp) activity and methyltransferase activity [97]. Subsequently, the negative-sense RNA that is produced is used to translate viral proteins at the endoplasmic reticulum and as a template for synthesizing new positive-sense genomic RNA by NS5 [90].

### 4.3. Assembly

As viral genomes and proteins are generated at the endoplasmic reticulum, ZIKV virions are assembled and trafficked to the Golgi apparatus for maturation [90].

### 4.4. Budding/Egress

Mature ZIKV particles are released from the cell via exocytosis from the Golgi apparatus [90]. Since the neurologic and teratogenic effect of ZIKV is of key concern, much research has been dedicated to the underlying mechanisms that drive these pathologies. Previously, the whole genome sequence of the ZIKV was isolated from the amniotic fluid of two pregnant women [98] and the fetal brain [99]. This highlights the tissue tropism of ZIKV infections toward neural tissues but also the ability of the virus to cross both the placental barrier and the blood–brain barrier (BBB) [100]. Although this ability to traverse the respective barriers remains largely unknown, recent cell culture models provide evidence to suggest that extracellular vesicles (EVs) containing tetraspanins may provide a platform for transport [100,101,102]. However, studies in animal models are yet to be undertaken. What is known from cell culture models is that EVs that are released from primary cultures of ZIKV-infected murine cortical neurons contained ZIKV RNA and proteins [102]. These EVs were found to be infectious and viable, even after treatment with RNaseA, Pitstop-2 (a clathrin inhibitor), or neutralizing antibodies that targeted the ZIKV envelope (E) protein. Viral loads of native ZIKV were hampered under the same treatments [102]. This suggests that EVs not only provide an alternate method for ZIKV infection, but infectious EVs are also capable of evading adaptive immune responses that depend on the recognition of the ZIKV E protein to function. Additionally, ZIKV infection in Vero E6 cells enhanced the production of EVs with sizes and densities that vary from EVs released from uninfected cells [101]. Since tetraspanins (CD9, CD63, and CD81) were shown to be critical in endosomal sorting complex required for transport (ESCRT)-dependent and -independent EV biogenesis, enhanced EV biogenesis was observed with a concomitant rise in CD63 levels 48 h post ZIKV infection in SNB-19 cells (human glioblastoma cells) [101]. This was accompanied by a simultaneous drop in ZIKV progeny virion release. Interestingly, when CD63 levels were ablated via shRNA (short-hairpin RNA) knockdown, the reverse was observed: release of ZIKV progeny virions was enhanced while infectious EVs correspondingly decreased [101]. Furthermore, ZIKV infection was associated with the localization of CD63 to perinuclear sites in the cell, where ZIKV replication occurs [101]. Since the release of infectious EVs may be modulated by the intracellular levels of tetraspanin CD63, the neurological and teratogenic effects elicited by ZIKV EVs may be controlled by limiting CD63 expression. Taken together, tetraspanins appear to not be involved in the entry and replication cycle of the ZIKV, but rather as part of the EV complex.

## 5. Influenza A Virus (IAV)

IAV is a highly pathogenic virus that causes seasonal flu every year, occasionally escalating to a pandemic. Most notably, it was the culprit for the Spanish flu in 1918 and, more recently, the H1N1 pandemic in 2009. Even with available vaccines and antiviral drugs, approximately 1 billion people contract an IAV infection every year, and up to 650,000 deaths are recorded globally [103]. Due to its highly mutagenic properties, IAVs undergo antigenic shifts to evade host immune responses, rendering flu vaccines ineffective after each season. IAV strains also develop resistance toward antiviral drugs, notably adamantanes (amantadine and rimantadine), which target the M2 ion channel protein to prevent replication within infected cells [104]. It is proposed that close to half of all globally circulating IAV subtypes are resistant to adamantanes, and the number is expected to continue to rise [104,105]. Thus, targeting conserved host factors that are critical for the life cycle of IAV provides an alternative or complementary strategy for developing novel antiviral drugs that are less prone to viral resistance. Tetraspanins are one of the novel host factors in IAV that could be therapeutically targeted.

### 5.1. Virus Entry

IAV is an enveloped, negative-sense single-stranded RNA virus whose outermost layer consists of three membrane proteins: hemagglutinin (HA), neuraminidase (NA), and M2 ion channel (M2). Beneath the viral envelope is a ring of matrix protein M1, which contains the eight segmented RNA genomes. IAV strains are named based on the HA and NA subtypes, and currently, 18 HA and 11 NA subtypes were described in influenza A [106]. To enter the host cells, the HA of the virus first binds to sialic acid receptors on the host cell surface glycoproteins, followed by endocytosis of the viral particles via a clathrin-dependent/independent pathway, or micropinocytosis [107,108,109,110] (Figure 4). Before viral uncoating, the virus is trafficked into the endosome, and the low pH condition inside the endosomes activates the M2 ion channels. Subsequently, this triggers the acidification of the inside of the virus and the first step of viral uncoating, which is characterized by the dissociation of the M1 matrix protein from the viral membrane [111,112]. The viral membrane can now fuse with the endosomal membrane, rendering the release of viral RNA genomes and their associated proteins into the cytoplasm. He et al. reported that CD81 is important for the fusion of the viral and endosomal membranes [113]. This was demonstrated by an IAV single-cell tracking experiment that showed that IAV is trafficked to CD81-positive endosomes before uncoating. Their results also showed a correlation between viral fusion within CD81-positive endosomes and infected cells expressing viral proteins, suggesting that CD81 organizes the endosomal membrane for viral fusion or directly trafficks IAV to endosomes for productive viral uncoating [113].

### 5.2. Nuclear Entry and Export

To enter the nucleus, IAV nucleoprotein (NP) expresses nuclear localization signals to enable it to be recognized by importin-α/importin-β heteromers, which are the host cell’s major cargo carrier of protein molecules from the cytoplasm to the nucleus [114]. Once inside the nucleus, the virus makes copies of itself before exiting the nucleus. Our group showed that CD151 is critical for the IAV life cycle as it mediates the export of virus progenies through its binding to newly synthesized viral proteins (NP, M1, and NEP) and the hosts’ nuclear export protein (CRM1) [115]. This is further supported by mouse data showing that CD151 knockout mice had significantly better survival rates than wild-type mice upon IAV infection, and knocking down the CD151 protein in IAV-infected patient-derived nasal epithelial cells significantly reduced the viral titers [115].

### 5.3. Virus Budding

Other than viral uncoating, CD81 also plays a role in viral protein assembly. Once new viral proteins and progenies are made, they are transported and assembled at the apical budding sites. He et al. showed that the IAV infection of A549 recruits CD81 on the plasma membrane to the concentrated budding sites where other viral proteins are located [113]. In support, CD81 knockdown prevented the budding virions from detaching off the plasma membrane. These observations suggest the CD81 may take part in the scission process that detaches the budding virions from the plasma membrane [113].

## 6. Coronavirus (CoV)

CoV is an enveloped, non-segmented positive-sense RNA virus that has the largest genome among the RNA viruses. In the past two decades, three major coronavirus outbreaks have occurred from the emergence of SARS-CoV-1 (2002–2003), MERS-CoV (2012), and SARS-CoV-2 (2019). Like most infectious diseases, the availability of vaccines is the most important aspect to prevent the spread of the virus. Although mRNA-based vaccines for SARS-CoV-2 have been widely utilized [116], the highly mutagenic and contagious nature of SARS-CoV-2 has made it difficult to contain. Therefore, it is critical to elucidate the replication mechanisms of CoVs to prevent potential future outbreaks.

The link between tetraspanins and CoVs is understudied due to there being only seven human CoVs that are emerging and/or circulating in the population. However, the emergence of the seventh CoV, namely, SARS-CoV-2, has caused renewed attention on host factors that interact with coronaviruses [117,118]. Part of this is due to the rapid emergence of SARS-CoV-2 variants that behaved differently to the parent strain, as well as between variants [119,120], necessitating the need to address gaps in understanding in variant–host interactions. Tetraspanins have been found to form membrane microdomains that facilitate the viral replication of many respiratory viruses. They are especially involved in the virus entry and budding of these viruses in their target epithelial cells [1,121]. Therefore, tetraspanin microdomains may serve as key interacting host factors in CoV replication cycles and, hence, may be a strong target for investigation, especially against SARS-CoV-2 and variants.

### 6.1. Virus Entry

CoV uses the tetraspanin microdomains for its entry and fusion alongside membrane-bound receptors and proteases within the microdomain to facilitate its receptor-mediated endocytosis entry and membrane fusion [122]. These microdomains, enriched with CD9, CD63, and CD81, were found to harbor CoV receptors and fusion activator proteases, arranging them in proximity to facilitate viral entry and fusion [122,123,124,125] (Figure 5). In the study by Earnest et al., a pulldown analysis of the CD9, CD63, and CD81 tetraspanin microdomains showed them to be harboring the CoV receptors APN (hCoV-229E), ACE2 (SARS-CoV and SARS-CoV-2), DPP4 (MERS-CoV), and CEACAM (mouse hepatitis virus, MHV), as well as the transmembrane serine protease TMPRSS2, which was utilized by coronaviruses for membrane fusion [125]. The study further showed that treatment with tetraspanin antibodies, anti-CD9, anti-CD63, and anti-CD81, reduced MHV infection, and SARS-CoV, MERS-CoV, and hCoV-229E VSV-pseudovirus transductions in susceptible cells [125]. Interestingly, antibody treatment of tetraspanins did not reduce the levels of CoV binding to the entry receptor, and MHV infection and VSV pseudovirus transduction were rescued with overexpression of TMPRSS2. The study hence confirmed that tetraspanin facilitation of CoV infections lies in mechanically allowing the access of receptor-bound viruses to the transmembrane proteases that speed up membrane fusion [125]. Moreover, another study by Earnest et al. further showed that CD9-enriched microdomains, but not CD81-enriched microdomains, were responsible for creating a DPP4–TMPRSS2 region for MERS-CoV infection, and knocking down either Cd9 or Tmprss2 (written in lowercase to distinguish it from the human gene) in mice resulted in reduced susceptibility in mice toward MERS-CoV infection [126]. More recently, aberrant expression levels of CD9 were also found in SARS-CoV-2-infected nasopharyngeal samples [127], which suggests the potential of CD9 being involved in facilitating SARS-CoV-2 entry and fusion via ACE2:TMPRSS2 microdomains. These findings further cement the vital role that tetraspanin microdomains play in CoV entry into host cells.

### 6.2. Transcription/Replication

So far, no association between tetraspanins and CoV transcription, protein synthesis, and replication has been identified. However, tetraspanins are known to be localized in different cellular compartments (nuclear/cytoplasmic membrane and endoplasmic reticulum, for example) and/or facilitate protein synthesis in the cells [128,129,130], albeit under other conditions, such as cancer and tumor formation. Therefore, there is a high likelihood that tetraspanins interact with the CoV genome/proteins during transcription and replication of the virus [131].

### 6.3. Assembly and 6.4 Budding/Egress

While it was established that tetraspanin microdomains are vital in CoV’s viral entry and fusion by putting the receptors and proteases in proximity, not much is known about its involvement in CoV assembly, budding, and egress. Tetraspanin CD81 is crucial in the viral protein packing, budding, and eventual egress for influenza viruses, but does not appear to aid in CoV egress [124]. However, this may also be due to the lack of detailed study in CoV budding and egress mechanisms, which was relatively understudied until the emergence of SARS-CoV-2. Given the current SARS-CoV-2-induced COVID-19 pandemic situation and the exponential research investigations in this area, there is no doubt that further studies detailing the contribution of tetraspanins in CoV signaling are expected.

## 7. Utility of Tetraspanins in Viral Disease and Future Outlook as a Target for Viral Infection

The data presented in the preceding sections demonstrate the critical and expansive role that tetraspanins play in virus infections (Table 1), all of which represent significant health and economic burdens worldwide. Furthermore, given the involvement of tetraspanins in the host cell machinery, the utility of targeting tetraspanins in viral infection is all the more enticing, especially with the emergence of novel viruses, the failure to develop efficacious vaccines (such as with HIV), and the limited utility of vaccines in diseases like influenza (due to the emergence of novel strains after each season) [132,133].

Across both infectious and chronic diseases, research on small molecule inhibitors targeting tetraspanins is only limited to HCV infections [2,11]. Small molecule inhibitors, as well as terfenadine and its derivatives, block the interaction between the CD81 large extracellular loop (LEL) domain and the HCV protein E2, thereby restricting viral entry into the host cell [11]. Separately, bis-imidazole-derived small molecules were synthesized to mimic the hydrophilic helix D region of CD81. These imidazole-derived small molecules serve as competitive inhibitors for the CD81 LEL-E2 protein interactions, inhibiting HCV entry [11]. However, it is unclear whether maximal efficacy can be achieved with small molecule inhibitors in vivo, as HCV studies to date are largely performed in cell cultures. In this review, HIV, IAV, and CoV viral proteins show direct interaction with tetraspanins, but the exact binding domains involved are unknown. Whether direct physical interaction between tetraspanins and viral proteins is necessary and sufficient for the life cycle of these viruses will need to be investigated if small molecule inhibitors are to be considered as treatment options for HIV, IAV, and CoV. Other approaches for targeting tetraspanins include monoclonal antibodies (mAbs) and gene deletion [2].

Antibody-mediated strategies primarily focus on targeting the physical interaction with tetraspanins. As described above, viruses are reliant on host cell mechanisms for various stages of their life cycle. Studies reporting the use of monoclonal antibodies that are specific to tetraspanins have concentrated on the viral entry stage. Treatment with anti-CD9, anti-CD63, and anti-CD81 mAbs was shown to inhibit coronaviruses at the cell entry stage and cause a reduction in MHV infection [125]. In contrast, anti-CD81 mAbs enhances HIV-1 Env-mediated viral entry in primary human T cells [56]. Therefore, using mAbs as a therapeutic in viral infections should be considered carefully, as they may have varying virus-specific consequences.

The majority of the studies that reported on tetraspanins utilized RNAi technology to inhibit the expression and function of tetraspanins. For in vitro experiments, RNAi methods include siRNA-, shRNA-, and CRISPR-Cas9-mediated knockdown of specific tetraspanins. At the virus entry stage, CD81 siRNA induced a defect in IAV fusion with the endosomal membrane [113], CD63 siRNA inhibited HIV-1 Env-mediated endocytosis [64], and CD151 siRNA reduced HPV16 endocytosis [22]. Of note, gene knockdown of these tetraspanins was correlated with reduced infectivity in their respective virus contexts. However, gene deletion may not be beneficial in a separate setting, including when CD9 and CD81 siRNA increased HIV-1 Env-mediated fusion and endocytosis [56], and CD9 siRNA enhanced HPV infection rates [24]. Upon nuclear entry, CD63 siRNA was shown to disrupt HIV-1 reverse transcription and integration into the host genome, which corresponded with a decrease in viral titer [74,75]. During viral assembly, CD81 siRNA prevented the detachment of budding virions from plasma membrane, which resulted in a reduction in progeny IAV [113]. Finally, CD151 siRNA was shown to inhibit the nuclear export of viral progeny and thus reduce IAV titers [115].

The remaining RNAi methods were used in ZIKV infection, where the ablation of CD63 in glioblastoma cells with shRNA resulted in the enhanced release of ZIKV progeny [101], and in coronavirus infection, where CRISPR-Cas9-mediated CD9 KO impaired MERS-CoV virus entry [126]. These findings, specifically those in which viral infection was impaired, demonstrate the potential of using RNAi technology to target tetraspanins in antiviral therapeutics.

An important step in progressing the development of these RNAi therapeutics is to investigate the impact of gene deletion in more physiologically representative culture models or in vivo. For instance, human nasal epithelial cells cultured in the air–liquid interface (ALI), which form a pseudostratified epithelium, were used to study IAV infection, as it better represented the biological features of the nasal epithelium compared to an immortalized monolayer cell culture [115]. While some tetraspanin knockout mice were shown to be viable [132], their uses in viral infection models are few. Only two studies related to the viruses included in this review have been reported: CD151 knockout mice, which exhibited a significant reduction in virus titers and improved survival upon IAV infection [115], and shRNA-mediated CD9 knockdown mice, which were less susceptible to MERS-CoV infection [126].

Despite these promising results, there are several limitations that should be noted for targeting tetraspanins in viral infections. These limitations can be subdivided according to virus-specific, host-cell-specific, and tetraspanin-specific limitations. First, regarding virus limitations, tetraspanin modulation does not consistently enhance or inhibit viral infection, even within the same system. One such example is HIV infection, where CD9 and CD81 were shown to decrease viral entry by reducing Env-mediated endocytosis [57,58]. However, an increase in progeny virus assembly was observed at TEMs containing these same tetraspanins [66,78,79]. Restricting inhibition of these tetraspanins to the budding/egress stage (and not the initial virus entry stage, as tetraspanin inhibition here could exacerbate infection) poses a substantial practical obstacle for the development of therapeutics. Second, the effect of tetraspanin modulation on the viral infection of host cells is not universal. For example, CD9 depletion decreased the HPV16 infection rate in HeLa and NHEK cells but enhanced infection in HaCaT cells [24]. In this case, targeted drug delivery to specific tissues or cells would be necessary to fully realize the benefit of tetraspanin inhibition in preventing viral infection.

Third, the nature of tetraspanins poses several limitations in developing therapeutics. As described above, therapeutic strategies that frame tetraspanins as therapeutic targets invariably aim to inhibit tetraspanin expression and function [10,132,133]. However, tetraspanins may play a beneficial role in preventing viral infection. For example, in HIV-1 infection, CD9, CD81, and CD63 inhibit both endocytosis [57,64] and virological synapse formation [66]. Inducing the expression of tetraspanins may be possible in vitro with overexpression vectors, but it is not as straightforward in a clinical setting, notwithstanding the added complication of the potentially unwanted detrimental effects of tetraspanin overexpression. Next, members of the tetraspanin family are closely related and often several tetraspanins are implicated in the same disease context. As discussed in the HIV section, redundancies and compensatory mechanisms in tetraspanin functions [80] may require a more extensive suppression of tetraspanins. One strategy to circumvent this limitation is to disrupt the TEM [132] via the targeting of tetraspanin palmitoylation, which was shown to play a critical role in TEM assembly [134]. However, there is a gap in the literature regarding methods of TEM disruption and their subsequent effect. Finally, another point of consideration is the widespread expression of tetraspanins under normal conditions and their physiological function. As an example, CD151 ablation was shown to be beneficial in treating IAV infection [115]. However, CD151 is ubiquitously expressed in normal human tissue [135] and plays key roles in several developmental and physiological processes, including kidney and skin development [136,137] and regulation of mast cell activation and T cell proliferation [138]. Similarly, CD9 influences cellular processes in different subtypes of immune and endothelial cells [139]. CD81 is heavily implicated in B lymphocyte development and activation, and CD63 is involved in cell migration and immune cell stimulation [140]. It is unknown whether the viruses that are documented in this review directly elevate the expression of tetraspanins. Still, there is the potential that viral infections can exacerbate disease conditions where the expression levels of tetraspanins are elevated compared to healthy controls. For example, viral respiratory infections in cancer patients are common and can worsen disease conditions and disrupt treatment [141,142,143,144]. The molecular mechanisms remain to be fully explored, but it is interesting to note that the expression levels of tetraspanin, CD151, CD9, CD81, and CD63 are elevated in disease conditions such as cancer [145,146,147]. Hence, the treatment of certain viral infections with tetraspanin inhibitors may be beneficial in cases where the virus depends on tetraspanins for its life cycle and the elevated expression level of the tetraspanin in question is characteristic of the disease severity.

## 8. Concluding Remarks

In conclusion, our review supports the study of tetraspanins regarding coronavirus, IAV, HIV, and HPV infections. There appear to be multiple tetraspanins that are involved at various stages of these virus’s life cycles, with CD9, CD61, CD81, and CD151 setting the scene, although to varying degrees of importance across the viruses discussed. Research into tetraspanins in viral infection is only just starting. Future directions on developing therapies that are targeted at tetraspanins will require in-depth knowledge of the molecular mechanisms of tetraspanins in viral infection to ensure that the physiological functions of tetraspanins in health and disease should also be balanced and devoid of adverse, off-target effects that are tailored to the virus in question.

## Figures and Tables

**Figure 1 ijms-22-11609-f001:**
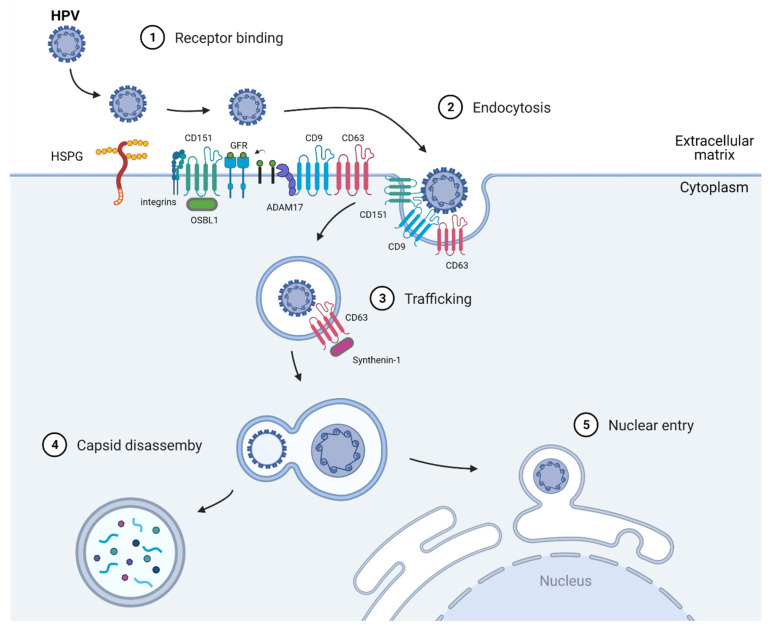
Diagram showing the contribution of tetraspanins in the HPV-1 replication cycle. CD151 and CD9 mediate HPV endocytosis in the secondary binding complex, together with adaptor proteins OBSL-1 and ADAM17, respectively. Meanwhile, CD63 and adaptor protein syntenin-1 are involved in the trafficking of the endocytosed HPV virus to multivesicular endosomes. Abbreviations: HPV, human papillomavirus; CD9, cluster of differentiation 9; CD63, cluster of differentiation 63; CD151, cluster of differentiation 151.

**Figure 2 ijms-22-11609-f002:**
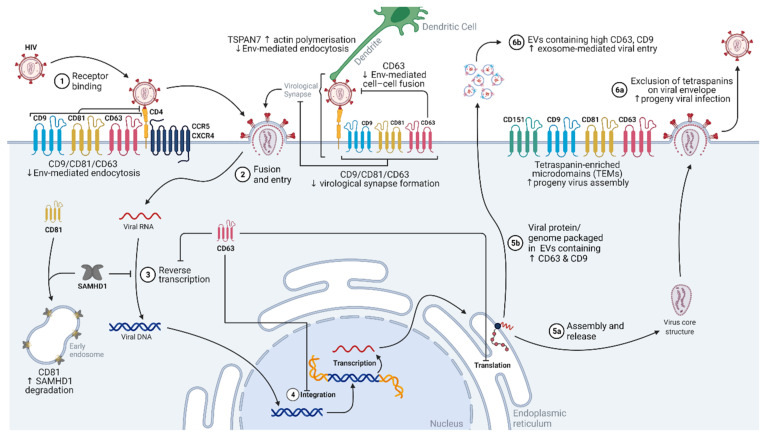
Diagram showing the contribution of tetraspanins in the HIV-1 replication cycle. CD9, CD81, and CD63 suppress HIV-1 viral entry, either through canonical endocytosis or the virological synapse. TSPAN7 also suppresses HIV-1 entry into dendritic cells by enhancing intracellular actin polymerization. Within the cell, CD81 binds to and promotes the degradation of SAMHD1, enhancing reverse transcription. CD63 is also associated with the inhibition of HIV-1 reverse transcription, integration, and translation. Tetraspanin-enriched microdomains (TEMs) serve as anchor sites for HIV-1 progeny assembly, while their integration in viral progeny envelopes is excluded. EVs with higher expressions of CD63 and CD9 contain the HIV-1 genome and/or viral proteins, and these EVs also have enhanced entries into target cells. Abbreviations: HIV, human immunodeficiency virus; CCR5, C-C chemokine receptor type 5; CD4, cluster of differentiation 4; CD9, cluster of differentiation 9; CD63, cluster of differentiation 63; CD81, cluster of differentiation 81; CD151, cluster of differentiation 151; CXCR-4, C-X-C chemokine receptor type 4; Env, envelope protein; EVs, extracellular vesicles; SAMHD1, sterile alpha motif (SAM) domain and histidine-aspartic domain (HD)-containing protein 1; TSPAN7, tetraspanin 7; vRNP, viral ribonucleoprotein.

**Figure 3 ijms-22-11609-f003:**
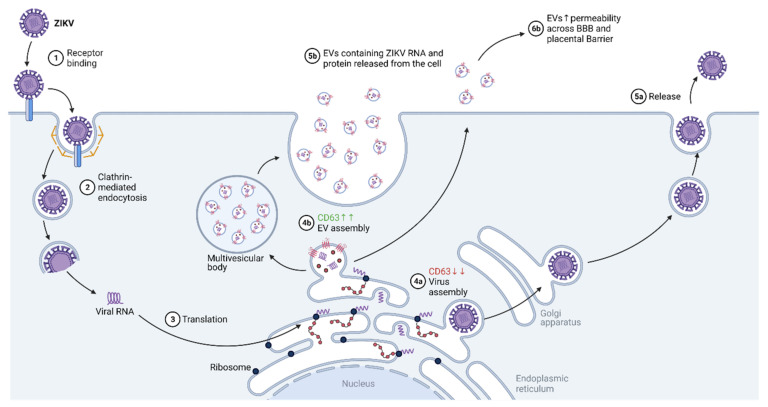
Diagram showing the contribution of tetraspanins in the ZIKV replication cycle. When the CD63 expression level is high, EV biosynthesis and release are enhanced. Conversely, when the CD63 expression level is low, ZIKV replication favors the formation of viral progeny. Abbreviations: ZIKV, Zika virus; BBB, blood–brain barrier; CD63, cluster of differentiation 63.

**Figure 4 ijms-22-11609-f004:**
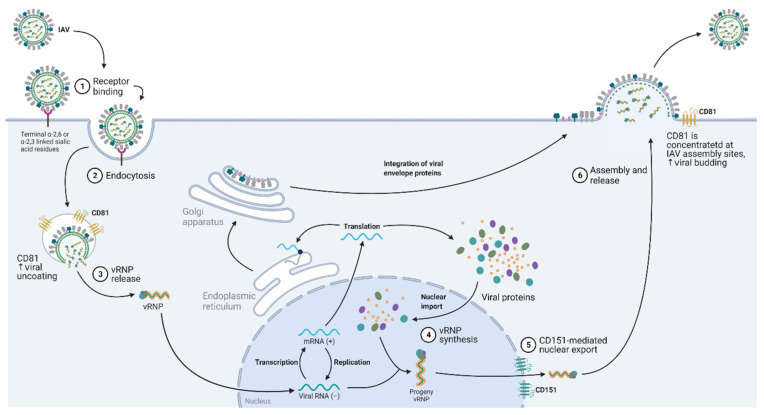
Diagram showing the contribution of tetraspanins in the influenza A virus replication cycle. Upon endocytosis, CD81 enhances viral uncoating, while CD151 facilitates vRNP nuclear export signaling. Abbreviations: IAV, influenza A virus; CD81, cluster of differentiation 81; CD151, cluster of differentiation 151; vRNP, viral ribonucleoprotein.

**Figure 5 ijms-22-11609-f005:**
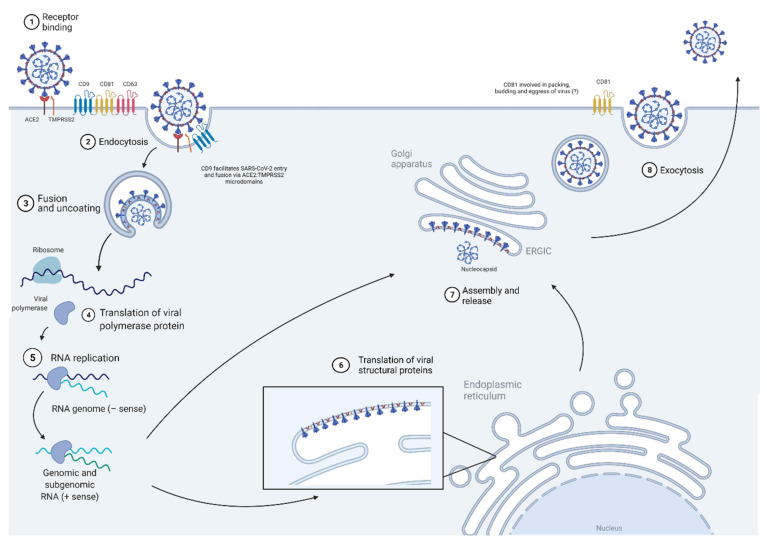
Diagram showing the contribution of tetraspanins in the SARS-CoV-2 replication cycle. CoV uses tetraspanin microdomains that are enriched with CD9, CD81, and CD63 for its entry and fusion. CD9 mediates the interactions between protease TMPRSS2 and CoV receptors, which are DDP4 in MERS-CoV and ACE2 in SARS-CoV-2. Abbreviations: CoV, coronavirus; ACE2, angiotensin-converting enzyme 2 receptor; TMPRSS2, transmembrane protease serine 2; CD9, cluster of differentiation; CD63, cluster of differentiation 63; CD81, cluster of differentiation 81.

**Table 1 ijms-22-11609-t001:** Summary of tetraspanin functions in various virus life cycles.

Tetraspanin	Virus	Role of Tetraspanins In Virus’s Life Cycles		
		Viral Entry	Replication	Viral Exit
**CD151**	HPV	[21,22]	-	-
	IAV	-	[114]	-
**CD9**	HPV	[23]	-	-
	HIV	[55,56,57,65]	-	[79,86]
	CoV	[121,122,123,124,125,126]	-	-
**CD63**	HPV	-	[19,24]	-
	HIV	[61,62,63,65,66]	[63,72,73,74]	[79,82,85]
	Zika	-	-	[100]
	CoV	[121,123,124]	-	-
**CD81**	HIV	[55,56,58,59,60,65]	[71]	[65,78,79,81,86]
	IAV	[112]	-	[112]
	CoV	[121,122,123,124]	-	-
**TSPAN7**	HIV	[50]	-	-

Abbreviations: CoV, coronavirus; HIV, human immunodeficiency virus; HPV, human papillomavirus; IAV, influenza A virus; ZIKV, Zika virus; CD9, cluster of differentiation 9; CD63, cluster of differentiation 63; CD81, cluster of differentiation 81; CD151, cluster of differentiation 151; TSPAN7, tetraspanin 7.

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
