# Peer review of "Tetraspanins: Host Factors in Viral Infections"

_ijms, 2021, doi:10.3390/ijms222111609_

Round 1

Reviewer 1 Report

This is an interesting review on the potential role of tetraspanins in the virus infections and it includes information not only on well-known viruses like HIV or IAV but also on the family of Coronaviruses to which SARS-CoV-2 belongs.

The work could be accepted after major revisions.

First English should be checked in any part of the manuscript as I found some sentences with mistakes or anyway in need of better clarification.

Section 2.1:

HPV is a small (approximately 8000 base pairs) ....virus. Please render better: virus genome is 8000 base pairs. 

The following frame should be rewritten in a clearer form 'In support, by comparing control and CD151 siRNA-treated HeLa cells incubated with HPV16 pseudovirions (PsV), they showed depleted post-internalized disassembled capsid specific antibody, L1-7 in the CD151 siRNA-treated cells but the level of surface capsid specific antibody, K75 remains the same between control and CD151 siRNA treated cells [17]ì

Fig.1-5: in all fig. legends please use Scheme in place of Schematic

sec. 2.2 Rewrite: ...showed that CD63 complexes with its cytoplasmic partner, syntenin-1, to aid in post-endocytosis trafficking....

section 2.3:

Rewrite: '...host cells’ cell cycle...'; 'HPV replicates themselves'; 

lines 127-131 '3. Discussion Authors should discuss the results and how they can be interpreted from the perspective of previous studies and of the working hypotheses. The findings and their implications should be discussed in the broadest context possible. Future research directions may also be highlighted.' should be deleted!

Sec 3.2: 'The viral DNA moves into the nuclease' may be into the nucleus?

Sec 3.3: Rewrite: 'free of Rab3a binding'

Sec 4.4: '6. Patents: This section is not mandatory but may be added if there are patents resulting from the work reported in this manuscript' should be deleted! 

Sec 5.2: rewrite: 'it to recruit importin-a, which the importin-ß receptor will then recognize..'

Sec 6: introduce better coronaviruses and SARS-CoV-2 and also report information on vaccines available also for these viruses as you reported for other viruses treated in the same manuscript. Cite at least work with DOI: DOI : 10.2174/0929867328666210521164809

The tetraspanin TSPAN8 was recently recognized as a mediator for SARS-CoV-2 infection. Please cite the work with DOI: 10.1101/2021.06.01.446640

Cd9 Tmprss2 and Covid-19 should be always be as capital letters

Reviewer 2 Report

This review by New et al. provides a detailed summary of the role of tetraspanins during viral infections. The authors describe the processes of host cell entry, replication, and egress of HPV, HIV, ZIKV, IAV, and CoV, and explain which tetraspanins are involved in each case.

Strengths of this study include the impressive amount of detail that is provided to the reader while maintaining a high degree of readability.

I do have a few questions and comments for the authors, which are listed below.

Major comments:

The authors chose to discuss the role of tetraspanins and their potential as targets for therapeutic intervention by detailing their role during infection with 5 viruses, HPV, HIV, ZIKV, IAV, and CoV. From literature, it appears that antiviral therapies targeting tetraspanins (mainly CD81), are described in most detail for hepatitis C virus (HCV; PMID: 24553110, 23704981). Is there any reason HCV was not included in this review?

In “Section 7: Utility of tetraspanins in viral disease and future outlook as a target for viral infection”, the authors outline monoclonal antibodies and RNAi as potential strategies for blocking tetraspanin expression/function. The use of small-molecule inhibitors could also be mentioned here, see another review on tetraspanins (PMID: 33137483).

Minor textual revisions:

Line 23: “multiple states of infectivity”: consider revising to: “multiple stages of infection”.

Line 54: throughout the term “coronavirus” is used. Often, but not always (such as here), is it clear whether the authors mean SARS-CoV-2 or the entire family of coronaviruses.

Line 62: “the underlying molecular mechanisms”: It is written as if it refers to the mechanism of action of HPV vaccines, that were mentioned immediately prior, but should refer to mechanism of infection. Consider rephrasing: “the underlying molecular mechanisms of infection”.

Line 77: in vitro, please use italics.

Lines 122-124: In the first of these two sentences, HPV is used as singular, but in the next sentence it is used as plural. Please check for consistency, I would recommend use as singular.

Lines 127-131: “3. Discussion Authors should discuss the results and how they can be interpreted from the perspective of previous studies and of the working hypotheses. The findings and their implications should be discussed in the broadest context possible. Future research directions may also be highlighted.”: This seems to be a piece of leftover text from an instruction resource for authors. Please remove.

Line 165: Please define “DCs” upon first use. For instance, introduce the acronym in line 138, when dendritic cells are first mentioned.

Line 399: “6. Patents This section is not mandatory but may be added if there are patents resulting from the work reported in this manuscript.” Again leftover author instructions, please remove.

Line 475: CoV is introduced in line 463 as plural (coronaviruses), but now used in singular. Please correct.

Line 493: “MERS-CoV-2”, I assume the authors meant MERS-CoV.

Reviewer 3 Report

ChihSheng et al aims to compare and contrast the role of tetraspanins in the life cycle of HPV, HIV, Zika, IAV, and coronavirus. These viruses cause the most significant health and economic burden to society. The manuscript is well written, should be of great interest to the readers. There are some concerns need to be further addressed:

1) In the introduction part, the authors introduced tetraspanins. It's too briefly. It should add more detailed information such as how many tetraspanin family members are there? What's the well-known members? What's their function in the human cells.

2) Since the review aim to compare and contrast the role of tetraspanins in different virus. It would be better to have a table to summary the roles of the same tetraspanins in different virus. 

3) Are there any small molecules targeting tetraspanins. If yes, the authors should add at least one paragraph to talk about these inhibitors.

Round 2

Reviewer 1 Report

The manuscript can be published in the current form